**Data Availability Statement:** All relevant data and model codes are available on GitHub (https://github.com/SelinHulagu/Material-for-SCLCO).

**Funding:** The author(s) received no specific funding for this work.

# Integrating life cycle assessment into supply chain optimization

**Selin Hülagü** [ID]*, **Wout Dullaert**, **A. Sena Eruguz** [ID], **Reinout Heijungs** [ID], **Dirk Inghels** [ID]

Department of Operations Analytics, School of Business and Economics, Vrije Universiteit Amsterdam, Amsterdam, The Netherlands

☯ These authors contributed equally to this work.

* s.h.huelague@vu.nl

## Abstract

Integrating Supply Chain Optimization (SCO) with Life Cycle Assessment (LCA) is essential for creating supply chains that are both economically efficient and environmentally sustainable. While SCO focuses on optimizing network structures and decisions related to product and service delivery, LCA systematically assesses the environmental impacts across the entire supply chain. The existing literature treats SCO and LCA as separate, sequential steps, often leading to inconsistencies in scope and challenges in data transfer and rescaling. Our research presents a novel Supply Chain Life Cycle Optimization (SCLCO) model that integrates SCO and LCA. Our SCLCO model is based on LCA data structures, incorporates multi-time period, closed-loop SCO decisions (e.g. reverse chain management, inventory control, network design), and is capable of considering the three pillars of sustainability: environmental, economic, and social. It includes harmonizing principles, terminology, and notation, thereby bridging the gap between the SCO and LCA communities through a generalized formulation. Computational experiments on a selected SCO model from Operations Research literature validate the SCLCO and demonstrate its effectiveness in providing valuable insights to both SCO and LCA practitioners and researchers. The results emphasize that the simultaneous execution of SCO and LCA in SCLCO minimizes the risk of overlooking decision impacts and facilitates data transfer from existing LCA databases.

## Introduction

Supply chain network optimization plays an essential role in improving sustainability by influencing decisions about materials used, sourcing, and reuse and recycling opportunities. Bringing products and/or services to customers and taking them back is the typical focus of supply chain network optimization and management in Operations Research (OR) and Supply Chain Management (SCM). To quantify environmental impacts, the Life Cycle Assessment (LCA) approach is well established. LCA follows a methodology standardized by the International Organization for Standardization (ISO, [1]). It is typically performed at the product level when decisions such as where to source, where and how much to store, and how to move and manufacture products have been made (e.g. [2, 3]). Many alternative scenarios are possible, and

**Competing interests:** The authors have declared that no competing interests exist.

identifying the best decisions is the subject of supply chain network optimization (SCO). SCO covers a wide range of decisions, including network design, inventory control, and process design.

The typical approach to incorporating environmental concerns into SCO is to consider a single impact category and use only the environmental impacts of a few activities (e.g. GHG emissions from transportation and production) as inputs to the optimization problem [4–6]. This partial assessment cannot accurately or comprehensively quantify the overall environmental impact, as the supply chain activities and environmental impact categories considered are not necessarily exhaustive.

Several studies have used LCA to measure and improve the environmental impacts of supply chains for different industries, motivating its use by its reliability and accuracy (e.g. [7–10]). The vast majority of this literature uses LCA to quantify the life cycle emissions of selected activities and uses LCA outputs as inputs for a SCO problem (e.g. [11–15]). Such an approach can be considered an *incorporation* of LCA into SCO, rather than an integration, as the two methodologies are performed sequentially. Typically, an LCA software is used as a black-box approach in such studies. Tan [16], Heijungs [17] provide a rigorous foundation on the computational principles of LCA to enable a more fundamental understanding, but they do not incorporate the optimization of supply chain decisions.

There are only a few OR studies that attempt to *integrate* LCA and SCO. However, these studies focus on specific tactical supply chain decisions within existing small-scale supply chains (e.g. [18]), or technology selection in a small-scale production system (e.g. [19]). Even studies that consider a broader set of supply chain decisions (e.g. those involving reverse chain decisions such as [20]) still neglect strategic decisions (e.g. related to network design), inventory decisions in multi-time period settings, and address the environmental impacts only. While focusing on environmental impacts is crucial, it is also essential to consider economic (traditionally the core of SCO) and social impacts in achieving comprehensive sustainability, as emphasized by regulations (e.g. CSRD [21]) based on Environmental Social Governance (ESG) point of views. To the best of our knowledge, no paper has integrated LCA into a comprehensive supply chain optimization problem involving a variety of network design and planning decisions while simultaneously considering economic and social impacts.

This paper proposes a method for structurally integrating LCA and SCO that (i) prevents overlooking environmentally, economically, and socially relevant decisions or processes and reports all relevant emissions and resource consumptions (even in multifunctional settings where multiple inputs or outputs are involved), (ii) limits data requirements and potential errors by supporting data transfer from existing LCA reports and databases, and (iii) proposes a generalized formulation capable of bridging the gap between the LCA and SCM communities, making SCO accessible to an LCA audience and vice versa.

We achieve these by (i) explicitly adopting LCA best practices when defining the scope of the analysis (for supply chain activities, emissions, and resource consumption), (ii) using LCA matrix notation and data structures, and (iii) proposing a Mixed-Integer Linear Programming (MILP) formula based on the above LCA best practices, notations, and computational structures.

The remainder of the paper is structured as follows. Section *Related literature* focuses on the sustainable SCM literature, which assesses environmental impacts through LCA. Section *Methodology* presents our novel approach to integrating LCA and SCO. We solve our model with a commercial MILP solver. The computational results on problem instances from the OR/SCM literature [14] are discussed in Section *Computational experiments and discussion*. Finally, conclusions and future research directions are presented in Section *Conclusions*.

## Related literature

This section focuses on sustainable supply chain optimization studies and solution approaches in which environmental impacts are measured using the LCA methodology. For more general information on environmentally sustainable supply chain optimization, we refer the reader to the literature reviews [4, 5, 22].

Table 1 provides an overview of peer-reviewed studies on sustainable supply chain optimization using LCA. We classify studies based on the scope of their LCA (i.e. cradle-to-gate, gate-to-grave, cradle-to-gate), and the type of their supply chain (i.e., forward supply chain, reverse supply chain, or closed-loop supply chain). For forward supply chains, papers typically conduct a "cradle-to-gate" analysis, capturing environmental impacts from raw material extraction up to the factory gate or the consumer. Papers focusing on reverse supply chains (e.g. waste management systems) usually only quantify the environmental impacts of the End-of-Life (EoL) of products, which can be classified under "gate-to-grave" analysis (from consumption to EoL). Environmental impacts over the entire life cycle of a product (from raw material extraction to EoL) are analyzed using a "cradle-to-grave" analysis. Since closed-loop supply chains (CLSCs) affect the entire life cycle of a product, a cradle-to-grave analysis is

**Table 1. Sustainable SCO studies that use LCA.**

| Reference | Supply Chain | | | Mathematical Model | | | LCA | | Integration |
|---|---|---|---|---|---|---|---|---|---|
| | Type | Product | Time | Objective (s) | Decisions | Solution Methodology | Scope | Environmental impacts | |
| Bloemhof-Ruwaard et al. [23] | RSC | M | S | En | F | Exact | CGR | 6-indicators | No |
| Azapagic and Clift [19] | FSC | S | S | Ec, En | F | Exact | CGT | 7-impacts | Partial |
| Hugo and Pistikopoulos [24] | FSC | M | M | Ec, En | F, N | Exact | CGT | EI 99 score | No |
| Dehghanian and Mansour [25] | RSC | S | S | Ec, En, So | F, N | Metah. | CGR | EI 99 score | No |
| Santibañez-Aguilar et al. [26] | FSC | M | S | Ec, En | F, N, I | Exact | CGR | EI 99 score | No |
| Chaabane, Ramudhin, and Paquet [7] | CLSC | M | M | Ec, En | F, N, I | Exact | CGR | GHG | No |
| Yue, Kim, and You [27] | FSC | M | S | Ec, En | F, N, I | Exact | CGR | GHG | No |
| Pishvaee, Razmi, and Torabi [28] | CLSC | M | S | Ec, En, So | F, N | Math. | CGR | ReCiPe score | No |
| Santibañez-Aguilar et al. [29] | FSC | M | M | Ec, En, So | F, N, I | Exact | CGT | EI 99 score | No |
| Vadenbo, Hellweg, and Guillén-Gosálbez [30] | RSC | M | S | Ec, En | F | Exact | GTGR | ReCiPe score | No |
| Altmann [31] | FSC | M | M | Ec | F, N | Exact | CGT | GHG | No |
| Mota et al. [12] | CLSC | M | M | Ec, En, So | F, N, I | Exact | CGT | ReCiPe score | No |
| Steubing et al. [18] | FSC | M | M | En | F | Exact | CGT | GHG | Partial |
| Babazadeh et al. [32] | FSC | M | M | Ec, En | F, N, I | Exact | CGT | Unspecified | No |
| Mota et al. [8] | CLSC | M | M | Ec, En, So | F, N, I | Exact | CGR | ReCiPe score | No |
| Sahebjamnia, Fathollahi-Fard, and Hajiaghaei-Keshteli [9] | CLSC | M | S | Ec, En, So | F, N | Metah. | CGR | ReCiPe score | No |
| Rohmer, Gerdessen, and Claassen [13] | FSC | M | S | Ec, En, So | F, N | Exact | CGT | GHG, Water | No |
| Negri et al. [10] | FSC | M | S | Env | F, N | Exact | CGR | ReCiPe score | No |
| Tautenhain et al. [14] | CLSC | M | M | Ec, En, So | F, N, I | Math. | CGR | ReCiPe score | No |
| This paper | CLSC | M | M | Ec, En, So | F, N, I | Exact | CGR | Flexible- case: ReCiPe score | Full |

Type: Forward Supply Chain (FSC), Reversed Supply Chain (RSC), Closed-Loop Supply Chain (CLSC); Product: Single (S), Multi (M); Time: Single (S), Multi (M); Objective(s): Economic (Ec), Environmental (En), Social (So); Decisions: Flow Decision (F), Network Design (N), Inventory Decision (I); Solution Methodology: Metaheuristic (Metah.), Matheuristic (Math.); Scope: Cradle to Gate (CGT), Cradle to Grave (CGR), Gate to Grave (GTGR); Environmental Impacts: Weighted Eco Indicator 99 score (EI 99 score), Greenhouse gas emissions (GHG), Weighted ReCiPe score (ReCiPe score), Water use (Water)

appropriate. In terms of solution methodology, many of the studies listed in Table 1 use exact methods (e.g., mixed-integer programming, $\epsilon$ constraint, weighted sum). A few studies develop metaheuristic or matheuristic algorithms for cases considered intractable for exact methods.

Most of the reviewed papers in Table 1 adopt a sequential approach to 'partially include' rather than 'fully integrate' LCA into their SCO problems. First, they perform an LCA to make an environmental assessment of a limited set of supply chain alternatives (e.g. different suppliers, production sites, or technologies). Then, an SCO model is used to select the best supply chain alternatives according to some objective function(s). Such a sequential approach has a number of drawbacks. If the environmental impacts and the supply chain decisions are evaluated separately, there is a greater probability that the scope of the environmental assessment and the decisions will differ. This issue is exacerbated by the fact that LCA is environmentally driven and supply chain management is economically driven, an observation made by Blass and Corbett [6]. The potential difference in the scope of the LCA and the SCO can be observed in the literature. Although there seems to be a relationship between the type of supply chain and the scope of the LCA conducted, the literature suggests that the type of supply chain does not always determine the scope of the LCA. For example, in [12], the supply chain includes more life cycle stages than the LCA that was conducted. Conversely, in [26], the scope of the LCA is broader than the life cycle stages covered by the supply chain. Most of the studies in Table 1 ignore the environmental impacts of certain processes in the LCA part of the analysis, even though these processes are part of the decisions in the SCO part. For instance, the following processes were excluded from the LCA part (without providing any motivation): raw material acquisition (e.g. [8, 33]), entity installment/entity capacity (e.g. [7]), and storage (e.g. [29]), although they have alternatives and related decisions in the SCO part. When LCA and SCO are fully integrated, there is a single scope for assessment and optimization. This ensures that all environmentally relevant activities are part of the optimization. This is the case for the studies that partially integrate LCA and optimization problems (e.g. [18, 19]).

Even if all of the LCA processes and SCO activities are aligned in the sequential approach (e.g. [26]), there are more issues to be addressed. The sequential approach poses data transfer and rescaling challenges. It requires LCA results be expressed in terms of unit flows (e.g., emissions to produce 1 kg of product, emissions to acquire 1 kg of raw material). This is not always the case in LCA databases and requires an additional step to rescale the data. By using an integrated model, issues related to data transfer between the parts can be completely avoided. Adopting LCA terminology and data structure in the optimization problem allows all aspects and steps of LCA to be part of the optimization.

Furthermore, multifunctional processes (e.g. a process that has more than one useful output) require an approach to allocate the resulting emissions in the sequential approach. Although it is common for processes to have multiple outputs and multifunctionality is a well-studied topic in LCA [17, 34], very few papers in the SCO literature discuss how they treat multifunctional processes. For example, Rohmer, Gerdessen, and Claassen [13] allocate emissions for a multifunctional process based on the economic value of its outputs. Allocation requires an additional step or additional side constraints to be added to the optimization model. However, if the entire supply chain is defined according to the LCA data structure, multifunctional processes and their emissions can be used as they are in linear programming settings.

A few papers have taken steps towards integrating LCA and optimization problems (e.g. [18–20]). Azapagic and Clift [19] use LCA methodology to represent a small-scale production system and Steubing et al. [18] use it to represent a small-scale supply chain. They focus on a limited set of supply chain decisions (e.g., flow decision, technology selection) in forward

supply chain settings. Freire, Thore, and Ferrao [20] include reverse supply chain decisions and provide a mathematical programming model based on activity analysis and LCA. Activity analysis is an established method for economic analysis and is similar to the LCA framework in its data structure and focus on individual activities (see [35]). Tan, Culaba, and Aviso [36] provide a fuzzy linear programming model based on LCA methodology and apply it for tactical decisions on power generation and biofuel generation systems. In the above studies, steps have been taken to partially integrate SCO and LCA. However, the scope of the decisions is limited to the tactical decisions and the scope of the LCA is limited to environmental impacts.

Technical and socio-economic criteria are integrated into the LCA-based optimization model of [18] for product design in [37]. Thies, Kieckhäfer, and Spengler [38] demonstrate how activity analysis and LCA integration can encompass economic and social pillars using an assessment framework. As such, these papers provide a valuable starting point for incorporating economic and social impacts into an LCA-based optimization framework.

Previous studies have attempted to combine SCO and LCA for (limited) tactical decisions, often using single-time-period models or focusing only on environmental impacts. A comprehensive integration of SCO and LCA that includes both strategic and tactical multi-time period CLSC decisions, while being able to consider economic and social impacts in addition to environmental assessment, is still lacking.

We present the first generic SCLCO model that integrates LCA and SCO decisions for multi-product, multi-time period CLSC problems. Our model addresses a comprehensive set of supply chain decisions, including reverse chain management, inventory control, and network design, while also considering the time-dependent relationships between these decisions. We also show a way of integrating economic and social impacts as focusing solely on environmental impacts may not always be sufficient or desirable in SCO. Our approach includes harmonization of principles, terminology, and notation. SCLCO provides convenience and avoids unnecessary allocation for multifunctional processes (e.g. joint production). The benefits we aim to achieve by integrating LCA and multi-time period, closed-loop SCO problems can be summarized as follows: (i) prevent overlooking the environmental quantification of any decision/process and report all emissions and resource consumption, (ii) consider processes and emissions as they are, even in multifunctional settings, (iii) be able to directly transfer data from LCA studies into the optimization model, (iv) bridge the LCA and SCM communities with a generalized formulation, (v) make SCO accessible to an LCA audience and vice versa.

## Methodology

Due to the multidisciplinary nature of our ambition to integrate LCA and SCO, the methodology section is presented in subsections aimed at establishing a common level of understanding before attempting the required integration. Section *Life cycle assessment (LCA)* explains the computational structure of LCA. To support this explanation, an illustrative example is provided in S1 File. Based on the LCA computational structure, the structure of an integrated generic SCO model for the three pillars of sustainability is proposed in Section *Supply chain life cycle optimization (SCLCO)*. For clarity, symbols commonly used in our SCLCO model are provided in the S1 File.

Our methodology is compatible with the international standards regarding LCA methodology (i.e., ISO 14040 series). Key components of our model align with these standards and are discussed in separate subsections, with supporting academic references: the matrix-data representation in Section *Organizing data*, the traditional LCA structure in Section *Model formulation*, the integration of existing LCA-based optimization models in Section *Supply chain life*

*cycle optimization (SCLCO)*, and the formulation of our final model in Section *SCLCO formulation*.

Throughout this paper, we use uppercase bold **X**, lowercase bold **x**, and uppercase/lowercase italic $X$/$x$ to represent matrices, column vectors, and scalars, respectively. The transpose of a vector or matrix is represented by the superscript **T**. **diag(x)** stands for a diagonal matrix constructed from a vector **x**. It has the values of **x** along its diagonal and zeros elsewhere.

## Life cycle assessment (LCA)

The ISO 14040 series provides a global standard for LCA methodology and guidelines for conducting an LCA. These standards define the phases of LCA [1, 39], establish a framework for data documentation [40], and provide examples of LCA applications [41, 42]. In this section, we follow the LCA methodology and key terminology as defined by ISO. However, we will also use more intuitive terminology to make the text easier to understand.

According to the LCA methodology, a product system is composed of processes (or unit processes). For each process, inputs and outputs are quantified. Each activity in a supply chain (e.g. sourcing raw materials, producing the final product) can be represented as a process. Every process absorbs or produces flows. These include goods (e.g. refined materials, intermediate/final products), services (e.g. transport), wastes, natural resources (e.g. water), and emissions (e.g. $CO_2$) [17]. There are two types of flows: product flows (i.e. exchanges between processes, also called intermediate flows in [1], intermediary flows in [43], economic flows in [17]) and environmental flows (i.e. exchanges between processes and the environment, also called elementary flows in [1]).

**Organizing data.** The calculation steps of LCA are well structured using a matrix-based representation (e.g. [16, 17, 43]), which is consistent with the data structure of LCA databases (e.g. [44]). This structure is also reminiscent of activity analysis (e.g. [20, 38]) and input-output analysis (e.g. [45–47]). In the matrix-based LCA structure, product flows are represented in the technology matrix **A**. **A** is an $N \times P$-dimensional matrix, where $N$ is the number of product flows and $P$ is the number of processes. Environmental flows are represented in the environmental intervention matrix $\mathbf{B^{env}}$. $\mathbf{B^{env}}$ is a $B \times P$-dimensional matrix, where $B$ is the number of environmental flows. In these matrices, each column corresponds to a process and each row corresponds to a flow. The elements in a matrix represent the absorption (negative value) and production (positive value) coefficients for the flows. Matrices **A** and $\mathbf{B^{env}}$ contain the absorption and production amounts of individual processes, per unit of production. The required performance of the system (required absorption and production amounts) is not specified in these individual recipes. The required performance of the system is defined by the "*functional unit*" (FU). The FU is the desired function(s) of the system(s) [48]. Defining the FU allows for a fair comparison between alternatives to achieve the same function [49]. Let the $N$-dimensional demand vector **f** consist of the demand coefficient for each product flow (row) in **A**. In general, **f** contains only a non-zero coefficient for the product flow corresponding to the reference flow [38].

**Model formulation.** To meet demand, each process must be scaled up (or down) by a certain amount. It is customary to assume a linear scaling of every process [50]. Given demand vector **f**, the scaling vector $\mathbf{s} \in \mathbb{R}^P$ can then be determined with Eq (1).

$$\mathbf{As} = \mathbf{f} \tag{1}$$

Eq (1) ensures that the system-wide aggregation of product flows (supply) is exactly the same as the demand vector under the conditions discussed in Section *Solving the model*.

System-wide aggregated environmental impacts are determined with the scaling vector, **s**. Environmental flows (represented in $\mathbf{B^{env}}$) can be aggregated into high-level impact categories (e.g. global warming, ecotoxicity) during the "impact assessment" phase of LCA, also known as LCIA [1]. In this phase, contributions of the environmental flows (e.g. $CO_2$, $CH_4$, $SO_2$) are converted into quantified environmental impact categories (e.g. global warming, acidification) based on the conversion factors defined by an impact assessment method (e.g. [51, 52]). This phase allows for both midpoint and endpoint methods. The characterization matrix $\mathbf{Q^{env}}$ includes the relative importance of the environmental flows (represented in $\mathbf{B^{env}}$) for the selected life cycle impact categories. $\mathbf{Q^{env}}$ is a $C \times B$-dimensional matrix, where $C$ is the number of impact categories. The entries in $\mathbf{Q^{env}}$ may vary depending on the impact assessment methodology, and the number of rows (which corresponds to the number of impact categories addressed) also varies per impact assessment methodology. Some studies define weighting factors to incorporate the relative importance of impact categories into the optimization problem (e.g. [51, 52]). Let the row vector $\mathbf{w^{env}} \in \mathbb{R}^C$ consist of weighting factors for each impact category. LCA result can be reduced to a single score, a weighted environmental index $W^{env}$, by Eq (2).

$$W^{env} = \mathbf{w^{env}}\mathbf{Q^{env}}\mathbf{B^{env}}\mathbf{s} \tag{2}$$

Breaking down Eq (2) we can clearly identify the three key LCA stages (defined by ISO [1]) from right to left, "inventory analysis" ($\mathbf{B^{env}s}$), "impact characterization" ($\mathbf{Q^{env}B^{env}s}$), and "weighting" ($\mathbf{w^{env}Q^{env}B^{env}s}$). We acknowledge the issue of weighting is topic of an ongoing debate (see, e.g., [53]). Nevertheless, given a vector of weighting factors, the proposed method works smoothly.

**Solving the model.** The scaling vector **s** from Eq (1) can be determined using traditional linear algebra methods. If matrix **A** is square (hence $N = P$) and non-singular, then from Eq (1) we obtain Eq (3):

$$\mathbf{s} = \mathbf{A^{-1}f} \tag{3}$$

where $\mathbf{A^{-1}}$ is the inverse of **A**. Notice from Eq (3) that if there is one non-zero element in the demand vector **f**, changes in **f** will have a linear proportional effect on **s**, and hence on $\mathbf{g^{env}}$, $\mathbf{h^{env}}$, and $W^{env}$ (e.g., doubling **f** will double $\mathbf{g^{env}}$, $\mathbf{h^{env}}$, and $W^{env}$).

Eq (3) is only valid if **A** is square. If there are alternative processes to produce a product, then **A** contains more columns than rows, $P > N$, making it not invertible. If $P > N$ then there are more unknowns than equations and the problem is underdetermined. This leads to an infinite number of solutions. In such cases, linear programming can identify the solution that gives the highest or lowest value to the objective function.

## Supply chain life cycle optimization (SCLCO)

In this section, we build on the LCA-based optimization models (e.g. [17, 18, 20]) to support multi-time period, multi-product, CLSC optimization decisions while considering environmental, economic, and social impacts.

Most of the sustainable SCO papers in the OR literature mainly focus on strategic decisions only (e.g. network design, technology installation, and truck purchase) [4]. There is also a body of research that combines strategic and tactical decisions (e.g. inventory planning and supply planning) [4]. By combining strategic and tactical decisions into a single model, the significant impact that different decision levels can have on each other can be captured [5]. We incorporate both strategic and tactical decisions while integrating SCO and LCA. Consistent with previous multi-time period SC research outlined in Table 1, we consider strategic

decisions to be long-term decisions (e.g. spanning several years) that are typically made only once in the scheduling horizon, whereas tactical decisions are short-term decisions (e.g. spanning months) that may be subject to adjustments.

CLSC optimization is a widely studied OR problem. It originally focused on minimizing costs and it increasingly focuses on minimizing environmental impacts (see Table 1). Traditionally, the problem is modeled using "flow-based" formulations (e.g. [7, 8, 54]), where decision variables represent product flows and switch on/off decisions (i.e. binary variables). The fundamental constraints of the problem are: (i) meeting customer/market demand, (ii) ensuring flow conservation between entities, and (iii) respecting maximum capacities [55]. We propose an alternative to the traditional flow-based SCO model based on the LCA data structure. Our model aims to find the best decisions that optimize overall environmental, economic, and social impacts. Unlike the traditional approach, the decision variables do not represent flows, but scaling factors (e.g. as in [18, 20]). In fact, impacts are not caused by product flows, but by the operation of their underlying processes [48].

The proposed SCLCO model is capable of handling the high-level characteristics of CLSC optimization models outlined in Table 1. It enables decision-making for both tactical and strategic CLSC decisions, including supplier selection and supply allocation, facility location, capacity determination, technology selection, production/remanufacturing planning, inventory planning, market allocation, and transportation planning. SCLCO allows for the optimization of individual or multiple products within the same supply chain. The ability to handle a large number of products within the same model is practical because processes can produce multiple products, and multiple products compete for the same process capacity. Moreover, SCLCO can accommodate multifunctional processes; a process can produce and treat multiple valuable products, by-products, and wastes (e.g., co-production, combined waste treatment).

**Extending standard LCA.** In order to formulate our integrated, multi-time period SCLCO model, featuring discrete and continuous variables for the strategic and tactical decisions common to CLSCs, some adaptations to the standard LCA procedure are needed.

Because our SCLCO distinguishes multiple time periods for which demand must be satisfied, we first introduce $t$ to indicate the time dimension for the demand vector $\mathbf{f}_t$ and the scaling vector $\mathbf{s}_t$. Multi-time period problem settings require flow connections between consecutive time periods. To facilitate this transition, we introduce a transition matrix $\mathbf{K}$ that maps the products from $t$ to $t + 1$.

Since our SCLCO includes both tactical and strategic variables, their environmental impacts must be quantified using the LCA data structure. The environmental impacts of tactical variables can be handled by standard LCA using (i) technology matrix $\mathbf{A}$, (ii) environmental intervention matrix $\mathbf{B}^{\text{env}}$, and (iii) scaling vector $\mathbf{s}_t$ for time $t$. We introduce a time dimension to $\mathbf{A}$ and $\mathbf{B}^{\text{env}}$ to capture changes in coefficients for product flows and environmental flows. For time periods $t = 1, \ldots, T$, $\mathbf{A}_t$ includes coefficients for product flows at time $t$ (e.g., the performance of a recycling center at each time period [7]), $\mathbf{B}_t^{\text{env}}$ includes coefficients for environmental flows at time $t$ (e.g. $CO_2$ producing MJ of electricity at each time period [56]), and $\mathbf{s}_t$ includes decisions at time $t$.

Using the standard LCA approach, the environmental impact of long-term (strategic) decisions, such as investments, can be handled by assuming a linear depreciation based on its expected lifetime (e.g. similar to renting a warehouse, where a monthly rent is charged rather than the total cost.). However, in SC network design, there are also (strategic) decisions for which existing SCO models take into account the full cost at a given point in time (e.g., building a facility, purchasing a fleet) (e.g. [8, 32, 57]), although such investments may also depreciated over time. We build our SCLCO in such a way that it is capable of handling both types of perspectives. For instance, all emissions from warehouse construction can also be accounted

for when making the network design decision, if the full cost of that decision is taken into account. This approach acknowledges that strategic decisions have long-term consequences and are fully responsible for the environmental impacts associated with those decisions. In this case, however, the linear scaling principle no longer applies because the emissions associated with the decision remain constant.

To express the environmental impact of strategic variables, we allocate the time period $t = 0$ to the product flows, environmental flows, and decisions related to the strategic processes and introduce the following elements: (i) strategic technology matrix $\mathbf{A}_0$, (ii) strategic environmental intervention matrix $\mathbf{B}_0^{\text{env}}$, and (iii) strategic scaling vector $\mathbf{s_0}$. $\mathbf{A}_0$ includes product flows that are absorbed/produced only once (e.g. construction materials for the construction of an entity). Similar to $\mathbf{A}_0$, $\mathbf{B}_0^{\text{env}}$ includes environmental flows that occur only once (e.g. emissions due to the construction of an entity). Vector $\mathbf{s}_0$ represents strategic decisions that are made once but affect the entire time horizon (e.g. construction of an entity).

The product flow from processes related to strategic decisions (e.g. warehouse construction) to the processes related to tactical decisions at each time period $t = 1, \ldots, T$ (e.g. warehouse space utilization) is represented in $\mathbf{A}'_t$. The overall technology matrix of the system at each time period can be depicted as:

$$\begin{bmatrix} \mathbf{A}_t & 0 \\ \mathbf{A}'_t & \mathbf{A}_0 \end{bmatrix} \quad t = 1, \ldots T$$

Although discrete variables are typically associated with strategic decisions, scaling factors can also be continuous or discrete, depending on the nature of the process. For example, it is possible to produce products in fractional quantities, such as half a kilogram, but it is not feasible to operate a fraction of a vehicle. Therefore, we extend the standard LCA to allow for discrete decision variables (e.g. integer).

**Organizing data for SCLCO.** We demonstrate the fundamental data structure of the SCLCO model using a hypothetical CLSC example. In this example, there is a single supplier that provides two types of raw materials that are co-produced at the same time, resulting in multifunctionality. To produce the final product, a technology (tch1 and/or tch2) must be installed at the plant and a warehouse with sufficient capacity must be built. The final product can either be remanufactured by the supplier to produce one of the raw materials, or it can be disposed of. Transportation between entities is facilitated by truck(s). Fig 1 illustrates the process-product flow diagram for this hypothetical example. In this example, there are $P^{\text{str}} = 4$ processes corresponding to the strategic decisions (in blue in Fig 1) and $P^{\text{tac}} = 9$ processes corresponding to the tactical decisions (in gray in Fig 1). There are $N^{\text{str}} = 4$ product flows produced by strategic processes (e.g. warehouse area) and $N^{\text{tac}} = 10$ product flows produced by tactical processes (e.g. final product at plant) in each time period.

We define the technology matrix of the system $\mathbf{A}_t$ for each time period $t = 0, \ldots, T$. For time period $t = 0$, $\mathbf{A}_0 \in \mathbb{R}^{N^{\text{str}} \times P^{\text{str}}}$ contains coefficients for the processes and product flows associated with strategic decisions. For time period $t = 1, \ldots, T$, $\mathbf{A}_t \in \mathbb{R}^{N^{\text{tac}} \times P^{\text{tac}}}$ contains coefficients for the processes and product flows associated with tactical decisions. The strategic product inflows to the tactical processes (e.g. tch1 capacity in Fig 1) are represented in $\mathbf{A}'_t \in \mathbb{R}^{N^{\text{str}} \times P^{\text{tac}}}$ $t = 1, \ldots, T$. The overall technology matrix for this hypothetical CLSC example for $t = 2$ is shown in Fig 2, which illustrates the construction of technology matrices and their interconnections. In Fig 2, the processes are labeled according to the numbering in Fig 1.

We define the environmental intervention matrix of the system $\mathbf{B}_t^{\text{env}}$ for each time period $t = 0, \ldots T$. For the time period $t = 0$, $\mathbf{B}_0^{\text{env}}$ includes environmental flows through the processes

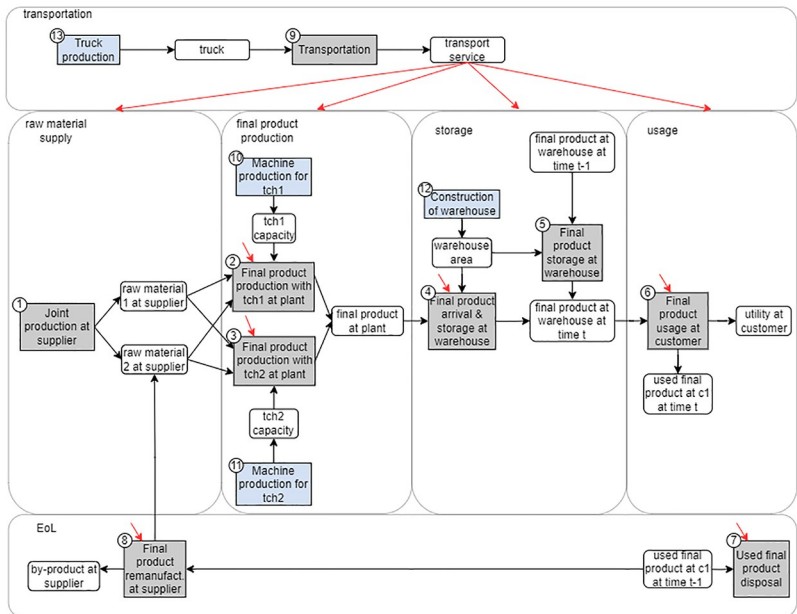

**Fig 1. Process-product diagram for the hypothetical CLSC example.** Rectangles represent processes (blue for strategic decisions, gray for tactical decisions) and rounded rectangles represent products. Red arrows represent transportation services.

corresponding to strategic decisions. For the time period $t = 1, \ldots T$, $\mathbf{B}_t^{\text{env}}$ includes environmental flows through the processes corresponding to tactical decisions. Given $B$ environmental flows, $\mathbf{B}_0^{\text{env}}$ and $\mathbf{B}_t^{\text{env}}$ are $B \times P^{\text{str}}$ and $B \times P^{\text{tac}}$ matrices, respectively.

The product transitions between consecutive time periods (e.g. used final product at customer at time t and time t-1) are facilitated by matrix **K**. This matrix is similar to the well-known Leslie matrices used in population growth studies to model the transition of individuals between different groups over time. In Leslie matrices, rows and columns correspond to stages, with entries indicating the probability of individuals transitioning to the next stage over time [58]. Our matrix **K** serves a similar purpose. It is structured as a binary square matrix, $\mathbf{K} \in \{0, 1\}^{N^{\text{tac}} \times N^{\text{tac}}}$. The columns correspond to products at time $t$, and the rows correspond to

|  | 1 | 2 | 3 | 4 | 5 | 6 | 7 | 8 | 9 | 10 | 11 | 12 | 13 |
|---|---|---|---|---|---|---|---|---|---|---|---|---|---|
| raw material 1 at supplier [kg] | 2 | -3 | -5 | 0 | 0 | 0 | 0 | 0 | 0 | 0 | 0 | 0 | 0 |
| raw material 2 at supplier [kg] | 8 | -2 | -6 | 0 | 0 | 0 | 0 | 1 | 0 | 0 | 0 | 0 | 0 |
| final product at plant [unit] | 0 | 2 | 3 | -1 | 0 | 0 | 0 | 0 | 0 | 0 | 0 | 0 | 0 |
| final product at warehouse at time t [unit] | 0 | 0 | 0 | 1 | 1 | -10 | 0 | 0 | 0 | 0 | 0 | 0 | 0 |
| final product at warehouse at time t-1 [unit] | 0 | 0 | 0 | 0 | -1 | 0 | 0 | 0 | 0 | 0 | 0 | 0 | 0 |
| utility at customer [kcal] | 0 | 0 | 0 | 0 | 0 | 50 | 0 | 0 | 0 | 0 | 0 | 0 | 0 |
| used final product at customer at time t [unit] | 0 | 0 | 0 | 0 | 0 | 10 | 0 | 0 | 0 | 0 | 0 | 0 | 0 |
| used final product at customer at time t-1 [unit] | 0 | 0 | 0 | 0 | 0 | 0 | -1 | -5 | 0 | 0 | 0 | 0 | 0 |
| by-product at supplier [kg] | 0 | 0 | 0 | 0 | 0 | 0 | 0 | 2 | 0 | 0 | 0 | 0 | 0 |
| transport service [kgkm] | 0 | -68 | -87 | -80 | 0 | -120 | -10 | -110 | 100 | 0 | 0 | 0 | 0 |
| tch1 capacity [final product] | 0 | -2 | 0 | 0 | 0 | 0 | 0 | 0 | 0 | 100 | 0 | 0 | 0 |
| tch2 capacity [final product] | 0 | 0 | -6 | 0 | 0 | 0 | 0 | 0 | 0 | 0 | 50 | 0 | 0 |
| warehouse area [m²] | 0 | 0 | 0 | -7.5 | -7.5 | 0 | 0 | 0 | 0 | 0 | 0 | 800 | 0 |
| truck [unit] | 0 | 0 | 0 | 0 | 0 | 0 | 0 | 0 | -0.5 | 0 | 0 | 0 | 1 |

**Fig 2. Technology matrices $\mathbf{A}_0$ (bottom right), $\mathbf{A}_{t=2}$ (top left), and $\mathbf{A}'_{t=2}$ (bottom left) and their connection at time period $t = 2$.**

$$
\mathbf{K}=
\begin{array}{r}
\text{raw material 1 at supplier [kg]} \\
\text{raw material 2 at supplier [kg]} \\
\text{final product at plant [unit]} \\
\text{final product at warehouse at time t [unit]} \\
\text{final product at warehouse at time t-1 [unit]} \\
\text{utility at customer [kcal]} \\
\text{used final product at customer at time t [unit]} \\
\text{used final product at customer at time t-1 [unit]} \\
\text{by-product at supplier [kg]} \\
\text{transport service [kgkm]}
\end{array}
\begin{bmatrix}
0 & 0 & 0 & 0 & 0 & 0 & 0 & 0 & 0 & 0 \\
0 & 0 & 0 & 0 & 0 & 0 & 0 & 0 & 0 & 0 \\
0 & 0 & 0 & 0 & 0 & 0 & 0 & 0 & 0 & 0 \\
0 & 0 & 0 & 0 & 0 & 0 & 0 & 0 & 0 & 0 \\
0 & 0 & 0 & 1 & 0 & 0 & 0 & 0 & 0 & 0 \\
0 & 0 & 0 & 0 & 0 & 0 & 0 & 0 & 0 & 0 \\
0 & 0 & 0 & 0 & 0 & 0 & 0 & 0 & 0 & 0 \\
0 & 0 & 0 & 0 & 0 & 0 & 1 & 0 & 0 & 0 \\
0 & 0 & 0 & 0 & 0 & 0 & 0 & 0 & 0 & 0 \\
0 & 0 & 0 & 0 & 0 & 0 & 0 & 0 & 0 & 0
\end{bmatrix}
\mathbf{e}=
\begin{bmatrix}
1 \\ 1 \\ 1 \\ 0 \\ 1 \\ 0 \\ 0 \\ 1 \\ 0 \\ 1
\end{bmatrix}
$$

**Fig 3. Matrix K and vector e for the hypothetical CLSC example.**

products at time $t + 1$. If there is a transfer from a product at time $t$ to a product at time $t + 1$, 1 is assigned, otherwise 0 is assigned. According to Fig 3, the final product at warehouse at time $t$ (column 4 of $\mathbf{K}$) is mapped in the consecutive time period to the final product at warehouse at time $t − 1$ (row 5 of $\mathbf{K}$), and the used product final product at customer at time $t$ is mapped in the consecutive time period to the used product final product at customer at time $t − 1$.

In this system, certain products may require complete processing within the system at each time period $t$ without any surplus (e.g. raw material 1 at supplier, final product at plant, final product at warehouse at time $t − 1$). To define which product flows are not allowed to have a surplus at time period $t$, we use the binary vector $\mathbf{e} \in \{0, 1\}^{N^{\text{tac}}}$. In vector $\mathbf{e}$, a value of 1 indicates a product flow that cannot have a surplus. Conversely, certain products have the potential for surplus at each time period $t$ (e.g. by-product at supplier, utility at customer, final product at warehouse at time $t$). Some of the surpluses are transferred to subsequent time periods (e.g., final product at warehouse at time $t$, used final product at customer at time $t$), while others are considered cut-offs and excluded from the analysis (e.g. by-product at supplier). Vector $(\mathbf{1} − \mathbf{e})$ (where $\mathbf{1}$ denotes $N^{\text{tac}}$-dimensional vector of ones) represents the product flows that are allowed to have a surplus at time period $t$. Fig 3 illustrates the $\mathbf{e}$ vector for the hypothetical CLSC example.

**Adding economic and social pillars.** In this subsection, we present an approach for integrating economic and social impacts into our optimization model, as focusing solely on environmental impacts may not always be sufficient or desirable. We are aware of the ongoing debates on assessing and integrating social impacts in optimization or assessment models (see [59, 60]), but consider them to be outside the scope of this study.

Following previous studies that handle the data structure of LCA (e.g. [38, 61]), we integrate economic and social flows in LCA in the same way that we represent environmental flows. $\mathbf{B}_0^{\text{ecn}}$ consists of economic flows that occur only once (e.g. the cost of constructing a factory). For time period $t = 1, \ldots, T$ $\mathbf{B}_t^{\text{ecn}}$ consists of economic flows that occur every time period (e.g. production costs, labor costs). There are $E$ economic flows that make $\mathbf{B}_0^{\text{ecn}} \in \mathbb{R}^{E \times P^{\text{str}}}$ and $\mathbf{B}_t^{\text{ecn}} \in \mathbb{R}^{E \times P^{\text{tac}}}$ $t = 1, \ldots, T$. Similarly, the intervention matrices $\mathbf{B}_0^{\text{soc}}$ and $\mathbf{B}_t^{\text{soc}}$ $t = 1, \ldots, T$ represent social flows that occur only once (e.g. number of workers for constructing a factory) and social flows that occur every time period (e.g. number of workers for production), respectively. Given $J$ social flows, $\mathbf{B}_0^{\text{soc}} \in \mathbb{R}^{J \times P^{\text{str}}}$ and $\mathbf{B}_t^{\text{soc}} \in \mathbb{R}^{J \times P^{\text{tac}}}$ $t = 1, \ldots, T$.

In the economic and social intervention matrices, undesirable flows are represented as outputs (positive values), while desirable flows are represented as inputs (negative values). For example, let's assume that the production cost per product is 10€ and the revenue per product

is 12€. In the economic intervention matrix, the cost would be represented as an output value of 10, while the revenue would be represented as an input value of -12. This representation allows the entire problem to be structured as a minimization problem.

We also define characterization matrices and weighting vectors for economic and social assessment. For $G$ economic impact categories, we define the economic characterization matrix as $\mathbf{Q}^{\mathbf{ecn}} \in \mathbb{R}^{G \times E}$ and the row vector $\mathbf{w}^{\mathbf{ecn}} \in \mathbb{R}^G$ consists of weighting factors for each economic impact category. For $L$ social impact categories, we define the social characterization matrix as $\mathbf{Q}^{\mathbf{soc}} \in \mathbb{R}^{L \times J}$ and the row vector $\mathbf{w}^{\mathbf{soc}} \in \mathbb{R}^L$ consists of weighting factors for each social impact category.

To consider the three pillars of sustainability together, we define new matrices. The intervention matrix $\mathbf{B}_t$ is a partitioned matrix covering environmental, economic, and social interventions. Matrices $\mathbf{Q}$ and $\mathbf{W}$ are defined to represent the characterization matrix and the impact category weights for the three pillars.

$$
\mathbf{B}_t = \begin{bmatrix} \mathbf{B}_t^{\mathbf{env}} \\ \mathbf{B}_t^{\mathbf{ecn}} \\ \mathbf{B}_t^{\mathbf{soc}} \end{bmatrix} \quad t = 0, \cdots, T, \quad \mathbf{Q} = \begin{bmatrix} \mathbf{Q}^{\mathbf{env}} & 0 & 0 \\ 0 & \mathbf{Q}^{\mathbf{ecn}} & 0 \\ 0 & 0 & \mathbf{Q}^{\mathbf{soc}} \end{bmatrix}, \quad \mathbf{W} = \begin{bmatrix} \mathbf{w}^{\mathbf{env}} & 0 & 0 \\ 0 & \mathbf{w}^{\mathbf{ecn}} & 0 \\ 0 & 0 & \mathbf{w}^{\mathbf{soc}} \end{bmatrix}
$$

**Decision variables.** We define scaling vectors $\mathbf{s}_t$ for each time period $t = 0, \ldots, T$. For time period $t = 0$, $\mathbf{s_0}$ is a $P^{\mathrm{str}}$-dimensional vector, consisting of strategic decisions. For time period $t = 1, \ldots, T$, $\mathbf{s}_t$ is associated with tactical decisions and indicates how much each process is performed at each time period, which is $P^{\mathrm{tac}}$-dimensional vector. Overall $\mathbf{s}_t$ $t = 0, \ldots, T$ defines the structure of the supply chain. It includes decisions on how, how much and where to produce (and remanufacture); how much and where to store and acquire; and how and how much to distribute.

**SCLCO formulation.** The overall Supply Chain Life Cycle Optimization (SCLCO) model is represented by (4).

$$
minimize \quad \sum_{t=0}^{T} \mathbf{W}\mathbf{Q}\mathbf{B}_t\mathbf{s}_t \tag{4a}
$$

$$
s.t.
$$

$$
\mathbf{diag}(\mathbf{e})(\mathbf{A}_t\mathbf{s}_t - \mathbf{f}_t) + \mathbf{K}\mathbf{A}_{t-1}\mathbf{s}_{t-1} = \mathbf{0}, \quad \forall t \in \{2, \cdots, T\} \tag{4b}
$$

$$
\mathbf{diag}(1 - \mathbf{e})(\mathbf{A}_t\mathbf{s}_t - \mathbf{f}_t) + \mathbf{K}\mathbf{A}_{t-1}\mathbf{s}_{t-1} \geq \mathbf{0}, \quad \forall t \in \{2, \cdots, T\} \tag{4c}
$$

$$
\mathbf{diag}(\mathbf{e})(\mathbf{A}_1\mathbf{s}_1 - \mathbf{f}_1) = \mathbf{0}, \tag{4d}
$$

$$
\mathbf{diag}(1 - \mathbf{e})(\mathbf{A}_1\mathbf{s}_1 - \mathbf{f}_1) \geq \mathbf{0}, \tag{4e}
$$

$$
\mathbf{A}_t'\mathbf{s}_t + \mathbf{A}_0\mathbf{s}_0 \geq \mathbf{0}, \quad \forall t \in \{1, \cdots, T\} \tag{4f}
$$

$$
\mathbf{A}_t^{\mathbf{out}}\mathbf{s}_t \leq \mathbf{u}_t, \quad \forall t \in \{0, \cdots, T\} \tag{4g}
$$

$$
\mathbf{s}_0 \in \mathbb{R}_+^{P_{\mathrm{con}}^{\mathrm{str}}} \times \mathbb{N}_0^{P_{\mathrm{dis}}^{\mathrm{str}}}, \quad \mathbf{s}_t \in \mathbb{R}_+^{P_{\mathrm{con}}^{\mathrm{tac}}} \times \mathbb{N}_0^{P_{\mathrm{dis}}^{\mathrm{tac}}}, \quad \forall t \in \{1, \cdots, T\}. \tag{4h}
$$

Solving the SCLCO problem boils down to finding the optimal scaling vectors for $P$ processes that minimize overall impacts. The multi-objective function is defined in Eq (4a). It includes separate environmental, economic, and social objectives. In other words, we do not prioritize one objective over the others. Eq (4a) is subject to the fundamental constraints of SCO problems: demand satisfaction, flow conservation, and capacity constraints. Demand satisfaction and flow conservation are guaranteed by Constraints (4b)–(4e). According to Eqs (4b) and (4c), the total product flow at time period $t$ ($\mathbf{A}_t\mathbf{s}_t$) together with the transferred product flows ($\mathbf{KA}_{t-1}\mathbf{s}_{t-1}$) from time period $t - 1$ must satisfy the demand at time $t$ ($\mathbf{f}_t$). Eqs (4a) and (4c) distinguish whether product flow surplus is allowed or not. Eq (4b) ensures that demand is satisfied without any product flow surplus. Eq (4c) allows for product flow surplus, which enables the product flows to exceed the demand vector. Initial demand is only satisfied by the initial product flows ($\mathbf{A}_1\mathbf{s}_1$) as ensured by constraints (4d) and (4e). In Eqs (4b)–(4e), $\mathbf{0}$ denotes the $N^{\mathrm{tac}}$-dimensional zero vector. Eqs (4b)–(4e) also ensure non-negativity in product flows. Product inflow and outflow requirements for processes related to strategic decisions are satisfied by Constraint (4f). In constraint (4f), $\mathbf{0}$ denotes the $N^{\mathrm{str}}$-dimensional zero vector. Constraint (4g) imposes an upper limit on the product outflows. In Eq (4g), for time period $t = 0$, $\mathbf{u}_0 \in \mathbb{R}^{N^{str}}$ contains the maximum outflow values for products related to strategic decisions. For time period $t = 1, \ldots, T$, $\mathbf{u}_t \in \mathbb{R}^{N^{tac}}$ contains the maximum outflow values for products related to tactical decisions. $\mathbf{A}_t^{\mathbf{out}}$ for $t = 0, \ldots, T$ is designed to contain only outflows. The domains of decision variables are given in Eq (4h). According to (4h), out of $P^{\mathrm{str}}$ strategic decisions, $P_{\mathrm{con}}^{\mathrm{str}}$ are continuous and $P_{\mathrm{dis}}^{\mathrm{str}}$ are discrete. Out of $P^{\mathrm{tac}}$ tactical decisions, $P_{\mathrm{con}}^{\mathrm{tac}}$ are continuous and $P_{\mathrm{dis}}^{\mathrm{tac}}$ are discrete.

## Computational experiments and discussion

In this section, we aim to validate the proposed SCLCO model and compare its effectiveness with a traditional SCO model to gain additional valuable insights, reduce the likelihood of overlooking relevant impacts, and facilitate the integration of LCA data for decision-making processes. To this end, we selected the SCO model and case presented by Tautenhain et al. [14] because it; (i) includes all the key strategic and tactical decisions identified in the literature (see Section *Related literature*, and Section *Supply chain life cycle optimization (SCLCO)*), (ii) incorporates the three dimensions of sustainability, and (iii) includes LCA results in its environmental objective. The selected case, mirroring a real-world electronic components supply chain, incorporates case-specific constraints (e.g., technology selection, minimum return fraction) and network structures. The computational experiments will demonstrate that using the SCLCO model yields results consistent with the original SCO model, confirming that our model can be used as an effective alternative. All the data used and the model codes are fully provided and digitally available on GitHub [62].

In the case [14], a single supplier (S) provides two types of raw materials (RM1 and RM2) for the production of a single type of final product (FP). The final product is manufactured in a single factory (F) using one of the three production technologies (TCH1, TCH2, TCH3). The factory has three alternatives for remanufacturing technologies (TCH4, TCH5, TCH6) for treating used final products (UFP). Final products can be stored in the factory warehouse (FW) or in two separate warehouses (W1 and W2). Two customers (C1 and C2) demand the final product. Customer C2 and warehouse W2 are located on a different continent from the other entities. Therefore, transportation to and from C2 and W2 is limited to air or sea transport. Each continent has one seaport (SEA1 and SEA2) and one airport (AIR1 and AIR2). There are two types of trucks for land transport (T1 and T2), one type of ship (SHP) for sea

transport, and one type of plane (PLN) for air transport. The time horizon is limited to two time periods.

The functional unit of the system is not explicitly defined in [14]. We propose defining the functional unit as "providing the customer with the dedicated desired utility at each time period". We assume that the desired utility is aligned with the customer demand for the final product. Therefore, the reference flow is set to the requirements provided by [14].

The data in the case, as reported in [14], is structured to be consistent with the traditional flow-based SCO models. Therefore, we present the data according to the LCA data structure adopted in this paper.

We use the provided data to organize the technology matrices ($\mathbf{A}_t$), the demand vector for each time period ($\mathbf{f}_t$), and the intervention matrices (e.g. $\mathbf{B}_t^{\mathbf{env}}$). In this case, $P$ is 179 ($P^{tac}$ = 142, $P^{str}$ = 37), $N$ is 134 ($N^{tac}$ = 81, $N^{str}$ = 53), $C$ is 17, and $G$ and $L$ are 1. All of the data to complete technology matrices $\mathbf{A}_t$ for $t = 0, \ldots, T$ and $\mathbf{A}'_t$ for $t = 1, \ldots, T$ is available in [14]. In this case, there are 3 distinct sets of technology matrices: $\mathbf{A}_0$, $\mathbf{A}_1 = \mathbf{A}_t\ t = 2, \ldots, T$, and $\mathbf{A}'_1 = \mathbf{A}'_t\ t = 2, \ldots, T$. These technology matrices are sparse matrices containing many zeros. They can easily be constructed using LCA software. Although there are many alternatives (e.g. CMLCA, SimaPro, GaBi), we chose to use CMLCA [63]. CMLCA is a free software tool that provides flexibility in its application because it does not impose predefined elements for allocation or impact assessment.

For each sustainability pillar, there are two distinct sets of intervention matrices corresponding to $t = 0$, and $t = 1, \ldots, T$. We complete the $\mathbf{Q}^{\mathbf{env}}\mathbf{B}_t^{\mathbf{env}}$ with the characterized environmental flows provided in [14]. In the original case study, economic flows are only expressed on monetary values. Social flows are represented by a social index based on Gross Domestic Product (GDP). Therefore, economic and social interventions are represented by the row vectors $\mathbf{b}_t^{\mathbf{ecn}}$ and $\mathbf{b}_t^{\mathbf{soc}}$. Environmental flows (ReCiPe endpoint) are considered to be undesirable, while economic (net present value) and social flows (GDP-index) are considered to be desirable, as explained in subsection.

While filling in the intervention vectors/matrices, we noticed that some data were missing. In some cases, the lack of data is related to interventions that were overlooked from a modeling perspective. These data gaps are discussed in Section *Reducing the chances of overlooking impacts*, but do not prevent us from comparing results with [14]. In other cases, it is clear that the data was not reported. We contacted the authors for the unreported data. In collaboration with the authors, we performed an in-depth verification of all input values used. Thanks to their support, we were able to update the dataset. The additional data can be found in S1 File. The data summarized in S1 File, together with the data reported in [14], constitute the updated dataset. The LCA data structure adoption of the updated dataset is available in [62]. Table 2 provides an overview of the data availability within [14]. In this table, a check mark (✓) indicates that the intervention data are available for the respective processes. A dash (–) indicates that the relevant intervention was ignored or overlooked for the respective processes. Data that was completed after contacting the authors of [14] are marked with a ×.

To fully reformulate the case study of [14], we incorporate additional case-specific features (e.g. minimum return fraction, investment budget) into our proposed SCLCO model, as described in S1 File. We implement the SCO model of [14] and the proposed SCLCO model using a computer with Intel(R) Core(TM) i7–1185G7, 3.00GHz processor and 16GB RAM. Both the SCO and SCLCO models are run individually for three objectives as reported in [14]: minimizing environmental impact, minimizing economic impact, and minimizing social impact. We obtain results for both models in less than one second. The proposed SCLCO model yields the same results for all objectives as the original SCO model for the updated

**Table 2. Data availability in [14] for our experiments.**

| Processes related to: | $\mathbf{Q}^{env} \mathbf{B}_t^{env}$ | $\mathbf{b}_t^{ecn}$ | $\mathbf{b}_t^{soc}$ |
|---|---|---|---|
| Raw material production | × | ✓ | – |
| Final product production | ✓ | ✓ | – |
| Product storage | – | ✓ | – |
| Product handling at airport/seaport | – | × | – |
| Product Usage | – | ✓ | – |
| Truck driving | – | × | – |
| Land/Air/Sea transportation | × | × | × |
| Planning truck | – | ✓ | ✓ |
| Constructing/Maintaining factory/warehouse | ✓ | ✓ | ✓ |
| Operating factory/warehouse | – | ✓ | ✓ |
| Operating airport/seaport | – | ✓ | – |
| Operating supplier | – | – | – |
| Installing technology | – | × | × |

dataset, which validates the proposed SCLCO formulation (see S1 File for more details on validation). The SCLCO model formulation, however, generates alternative model outputs that allow for a broader and easier interpretation, especially for practitioners and researchers with an LCA background. In the following sections, we examine the key decision variables and their associated information.

## Interpretting the solutions of SCLCO

Solving the SCLCO problem provides the optimal scaling vectors for strategic and tactical decisions. $\mathbf{s}_0$ represents scaling factors for processes related to the strategic decisions and determines fleet composition, entity construction, and, technology and supplier selection. $\mathbf{s}_t$ includes scaling factors for processes related to the tactical decisions and defines raw material acquisition amounts, production and remanufacturing quantities, storage levels, and transportation choices (e.g., air or sea). Unlike traditional SCO models that output explicit flows, our model implicitly defines flows through $\mathbf{s}_0$ and $\mathbf{s}_t$, as they represent the in- and outflows of the underlying processes. For the case under consideration, $\mathbf{s}_0 \in \mathbb{R}_+^3 \times \mathbb{N}_0^{34}$, $\mathbf{s}_t \in \mathbb{R}_+^{92} \times \mathbb{N}_0^{50}$. In these vectors, rows greater than zero indicate the processes used in the solution. Rows that are zero indicate that these processes are not involved in the solution structure. The scaling factors for the processes that are used in the solutions (greater than zero) are provided in S1 File and are available in [62].

The SCLCO model provides other valuable insights that can be useful to both SCO and LCA practitioners and researchers.

SCO studies (such as [14]) typically report the values of the key decision variables, which typically represent flows in the network. In SCLCO, the key decision variables are the scaling vectors for the processes. The flows are the inputs and outputs of the processes and they are determined by the scaling vectors. Product inflows/outflows to/from processes at each time period $t \in \{0, 1, 2\}$ can be obtained and reported by Eq (3).

$$\mathbf{A}_t \mathbf{diag}(\mathbf{s}_t) \tag{5}$$

In the literature, it is common to visualize the flows in the solution, especially for networks of a compatible size (e.g. [18, 24, 29, 30]). In Fig 4, we provide a visual representation of the flows of raw materials, final products, stored final products, and used final products within the

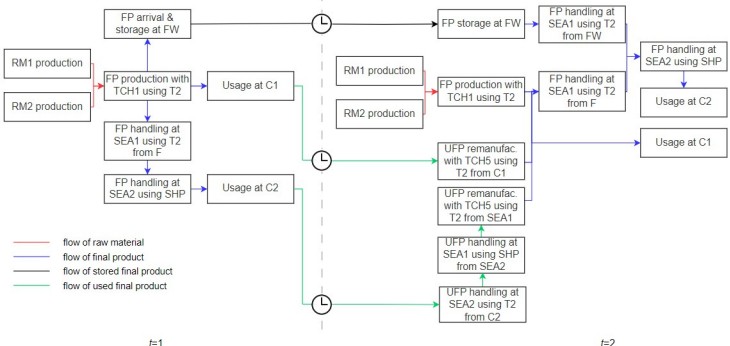

**Fig 4. Raw material, final product, stored final products, used final product flows in the environmental solution for $t$ = 1, 2.**

environmental solution for $t$ = 1, 2 (as in [14]). Our representation also illustrates the interactions between the processes.

Following LCA best practices, we conduct a contribution analysis. Analyzing the contribution of the decisions/activities/processes to the overall impact helps to identify the 'hot spots' in the supply chain, key areas for finding new alternatives. Such a contribution analysis can guide decision makers to further improve the sustainability of their supply chains. In the flow-based SCO model, determining the impact of each decision requires multiplying each decision individually by the corresponding impact parameter, which can be cumbersome when there are numerous decision variables involved. In the proposed SCLCO, the contribution analysis can be conducted simply by multiplying the intervention matrix/characterization matrix/weighed index by the diagonal matrix of the corresponding scaling vector. For instance, environmental impacts at each time period $t \in \{0, 1, 2\}$ can be decomposed into the processes by Eq (6).

$$\mathbf{w}^{\text{env}}\mathbf{Q}^{\text{env}}\mathbf{B}_t^{\text{env}}\mathbf{diag}(\mathbf{s}_t) \tag{6}$$

In Fig 5, the processes shown contribute more than 15% to the environmental, economic, and social impacts in the three SC configurations corresponding to the three objectives (min $W^{\text{env}}$, min $W^{\text{ecn}}$, min $W^{\text{soc}}$). The environmental impacts in the case under consideration are primarily driven by raw material production (RM1 production) and final product production

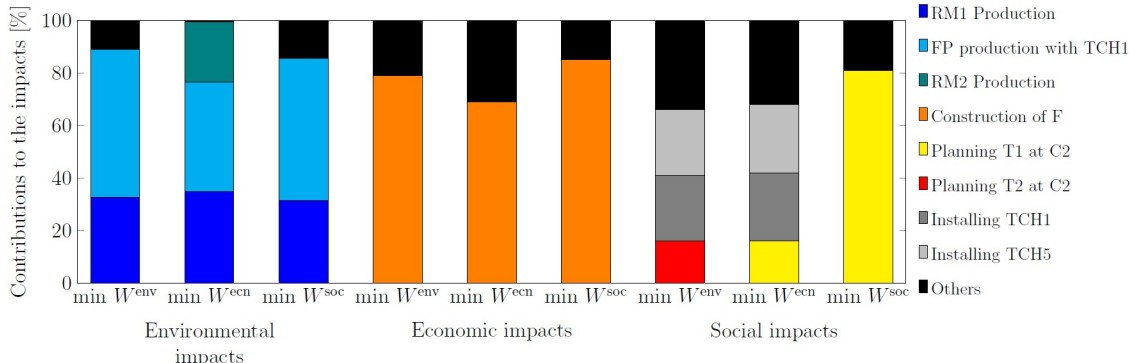

**Fig 5. Contributions of environmental, economic, and social impacts to environmental (min $W^{\text{env}}$), economic (min $W^{\text{ecn}}$), and social (min $W^{\text{soc}}$) objectives.**

(FP production with TCH1 at F) in all SC configurations. The selected technology in the economic solution increases the use of raw material 2 and therefore its contribution to the environmental objective. For the economic impacts (only the contribution of costs is shown), the majority of the costs, exceeding 69% in all configurations, are related to the construction of the factory (Construction of F). For the social impacts (measured by a GDP-based index), about half is attributed to technology installation (Installing TCH1 and Installing TCH5) in environmental and economic solutions. Social impacts in the social solution are mainly contributed by the truck purchased (Planning T1 at C2).

## Reducing the chances of overlooking impacts

In Section *Related literature*, we argued that the proposed SCLCO model reduces the chances of overlooking the impacts of decisions. In the traditional SCO model of [14], the impacts of some decisions (e.g. Table 2) were excluded. This can be seen by examining the intervention/impact vectors/matrices ($\mathbf{Q}^{\mathbf{env}}\mathbf{B}_t^{\mathbf{env}}$, $\mathbf{b}_t^{\mathbf{ecn}}$, $\mathbf{b}_t^{\mathbf{soc}}$) of the updated dataset in the SCLCO. In these matrices, each process is represented by a separate column to which a numerical value must be assigned. All processes that are intentionally (e.g. process with negligible interventions) or unintentionally (e.g. overlooked) excluded are assigned a value of zero. Table 2 outlines whether or not process interventions are reported.

For the environmental pillar, interventions for processes such as storage, product handling at seaports, construction of seaports, and usage are not reported. Economic interventions are explicitly or implicitly reported for almost all processes, with the exception of costs associated with operating a supplier location. Social impacts are affected by only a few activities such as entity construction, production/remanufacturing, and planning trucks.

Processes that do not consider interventions from any of the three pillars (e.g., operating supplier locations), can be considered outside the scope of the decision maker. However, when a process is considered from at least one pillar, it falls within the scope of the decision maker. Considering the economic interventions for almost all reported processes, the missing ones regarding environmental and social interventions can be seen as "overlooked". For a fair assessment and a comprehensive solution, we firmly believe that there needs to be alignment between the interventions considered for the three pillars. Ideally, the intervention matrices would be equally complete. Otherwise, there could be an imbalance in the evaluation, resulting in an incomplete understanding of the overall impact.

By requiring that data be filled in for each process, we believe that the SCLCO approach minimizes the risk of inadvertently overlooking processes. This approach also provides a clear visualization and identification of the decisions that have not been assessed. It thus provides a basis for discussing the reasons for their exclusion. When a particular case is examined, some impacts or even a particular sustainability pillar can be considered out of scope. However, we believe that it is necessary to consider the economic, environmental and social interventions of each decision (process). If these interventions are negligible or zero, it is reasonable to state this.

## Transferring data from LCA studies

In traditional SCO models, incorporating new data from LCA studies requires the introduction of new parameters. However, the proposed SCLCO model does not require any additional parameters, constraints, or decision variables. Therefore, data from LCA studies or LCA databases can easily be combined. This is because SCLCO is based on the LCA data structure. We fill the $\mathbf{B}_t^{\mathbf{env}}$ from LCA studies, focusing specifically on processes whose impacts are not environmentally quantified in [14]: storage process, product handling of incoming goods at

**Table 3. Absolute and relative changes in environmental, economic, and social impacts.**

|  | $\Delta W^{env}$ | $\Delta W^{ecn}$ | $\Delta W^{soc}$ | $\Delta_{\%} W^{env}$ | $\Delta_{\%} W^{ecn}$ | $\Delta_{\%} W^{soc}$ |
|---|---|---|---|---|---|---|
| min $W^{env}$ | 2722 | 28551 | -0.04 | 0.7% | 21.2% | -0.4% |
| min $W^{ecn}$ | 1295 | 0 | 0 | 0.2% | 0% | 0% |
| min $W^{soc}$ | 2061 | 0 | 0 | 0.5% | 0% | 0% |

seaports/airports (see details on data in S1 File). New data for the relevant processes are incorporated into the SCLCO model by filling in the corresponding column in the environmental intervention matrix. This allows the newly added impacts to be quantified in the objective function and their optimal values to be determined directly within the SCLCO model.

We re-optimize the case under consideration for the three sustainability pillars using the new data. The new environmental solution is different from the previous solution (Section *Interpretting the solutions of SCLCO*). While the first solution requires product storage in the factory warehouse, none of the products are stored in the new environmental solution. This new environmental structure significantly affects the economic impacts. For the economic and social pillars, the solution structures remain the same because no new economic and social data was added. Changes in environmental, economic, and social impacts are summarized in Table 3. In the new environmental solution, the economic impact (costs) increases by 21.2%. Although the changes in environmental impacts are relatively small (around 1%) for all solutions, it should be noted that in this case study the final product is an electronic component and the storage/handling processes for such products have limited environmental impacts. For other product types (e.g., fresh products), storage and handling can significantly influence the environmental impact (e.g., [64]). Our main argument is that the proposed SCLCO approach facilitates the incorporation of all environmental effects e.g., by relying on published LCA data.

## Conclusions

The optimization of supply chain networks has evolved to include consideration of environmental impacts. LCA is a well-established method for quantifying these impacts. Many SCO studies attempt to improve the environmental performance of supply chain networks by incorporating LCA features. In the existing literature, LCA is typically conducted as a separate step prior to SCO, which often leads to differences in scope and challenges in data transfer and rescaling. Only a few papers have attempted to integrate LCA and SCO for limited supply chain decisions (e.g. flow decision, technology selection). In this paper, we present a novel formulation that integrates LCA and SCO. Unlike previous studies attempting such integration, our proposed SCLCO model comprehensively addresses multi-time closed-loop SCO tactical and strategic decisions and explicitly considers the three pillars of sustainability. The proposed SCLCO model optimizes decisions related to closed-loop chain management, inventory control, and network design, taking into account their environmental, economic, and social impacts. Leveraging the LCA data structure, we extend the traditional LCA methodology to include strategic decisions, discrete variables, and multi-time period settings, effectively transforming the traditional SCO problem into an SCLCO model. The SCLCO model aims to propose a generic model formulation that improves the accessibility of LCA practitioners or researchers to SCM concepts and vice versa. Moreover, it opens new opportunities for collaboration and knowledge exchange between these two traditionally separate domains. As a result, it improves the overall understanding and application of sustainability principles in supply chain decision making.

Computational experiments on a case from OR/SCM literature (see previous formulation in [14]) not only validate our SCLCO, but also highlight its flexibility in handling case-specific features. The computational experiments yield three key findings, which are presented in this paper:

- The solution of the SCLCO model provides optimal scaling vectors for processes related to tactical and strategic decisions, providing valuable insights to both SCO and LCA practitioners and researchers. This streamlines post-analysis tasks, including contribution analysis, while still accommodating the needs of SCO practitioners and researchers, such as the use of flow diagrams.

- Moreover, the SCLCO model enables the simultaneous execution of LCA and SCO, thereby overcoming the scope difference limitation associated with the traditional sequential approach. We demonstrate how the SCLCO data structure enables clear visualization and identification of excluded processes/activities, thereby minimizing the risk of overlooking the impacts of decisions.

- Finally, the proposed SCLCO model illustrates the ease of data transfer from LCA studies by embracing the LCA data structure. We show that the optimal solutions may differ when transferring data for previously overlooked processes. For the case under consideration, the environmental, economic, and social impacts change between 1 and 21%.

One of the potential limitations of this study is that it is rooted in traditional, environmental LCA, therefore potentially excluding challenges arising from integrating SCO with less adopted approaches such as Social Life Cycle Assessment or Life Cycle Costing. We consider such an integration to be a promising area for future research. Future research can also focus on the practical implementation of SCLCO in real-world case studies. This will reveal how SCLCO can be adjusted to different case-specific parameters and constraints, further extending its practical applications (e.g. incorporating production and replenishment lead times). We also want to encourage researchers to use the framework to tackle large-scale real-world instances, which will present specific challenges to the model and solution approaches. In addition, it may be worthwhile to extend SCLCO to multi-objective settings and explore the development of tailored multi-objective optimization solution approaches. SCLCO can be extended to capture uncertainties (e.g. [65]), i.e. to make the model stochastic. The rich infrastructure available for LCA (e.g. software, databases, ISO standards, governmental policies) may be a good vehicle to implement SCLCO more easily and widely, and also to defend such efforts in negotiations with contractors and policy officials.

## Supporting information

**S1 File. Supplementary material.**
(PDF)

## Author Contributions

**Conceptualization:** Selin Hülagü, Wout Dullaert, A. Sena Eruguz, Reinout Heijungs, Dirk Inghels.

**Data curation:** Selin Hülagü.

**Formal analysis:** Selin Hülagü.

**Methodology:** Selin Hülagü, Wout Dullaert, A. Sena Eruguz, Reinout Heijungs, Dirk Inghels.

**Resources:** Selin Hülagü.

**Supervision:** Wout Dullaert, A. Sena Eruguz, Reinout Heijungs, Dirk Inghels.

**Validation:** Selin Hülagü.

**Visualization:** Selin Hülagü.

**Writing – original draft:** Selin Hülagü.

**Writing – review & editing:** Wout Dullaert, A. Sena Eruguz, Reinout Heijungs, Dirk Inghels.

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
