## [Decision Letter · Decision Letter 0]

17 Oct 2024

PONE-D-24-36569Integrating Life Cycle Assessment into Supply Chain OptimizationPLOS ONE

Dear Dr. Hulagu,

Thank you for submitting your manuscript to PLOS ONE. After careful consideration, we feel that it has merit but does not fully meet PLOS ONE’s publication criteria as it currently stands. Therefore, we invite you to submit a revised version of the manuscript that addresses the points raised during the review process.

Dear Authors,  Thank you for your submission to PLOS ONE. After careful and extensive review, I note that your submission requires major revisions before it can be accepted for publication. I invite you to consider all the review comments appended below and revise your manuscript to acceptable standards. I suggest to pay particular attention on LCA methodology and model validation as requested by Reviewer #1

We look forward to receiving your revised manuscript.

Kind regards,

Alberto Barbaresi

Academic Editor

PLOS ONE

Reviewers' comments:

Reviewer's Responses to Questions

**Comments to the Author**

1. Is the manuscript technically sound, and do the data support the conclusions?

Reviewer #1: Partly

Reviewer #2: Yes

2. Has the statistical analysis been performed appropriately and rigorously? 

Reviewer #1: N/A

Reviewer #2: N/A

3. Have the authors made all data underlying the findings in their manuscript fully available?

Reviewer #1: Yes

Reviewer #2: No

4. Is the manuscript presented in an intelligible fashion and written in standard English?

Reviewer #1: Yes

Reviewer #2: Yes

5. Review Comments to the Author

Reviewer #1: Dear authors,

the work is really interesting and innovative, which is properly stressed in the article. However, I think some aspects should be clarified before considering it for publication.

Please consider the comments below:

- Considering the number of acronyms, I suggest including a nomenclature section;

- The authors talk about integrating the LCA method at the level of supply chain optimization. An important feature of an LCA study is the definition of the impacts that a process can have on the environment. How is this aspect quantifiable in your integrated model? Do you foresee factors for characterizing the impacts and how are they integrated at the methodological level? In analogy to this aspect how is treated the stage of Environmental impact assessment?

- the methodology has been presented at the methodological and formulation level but it is not clear whether the model has ever been validated? Has it been tested on any real cases? This aspect is fundamental for the reliability and repeatability of the work.

- Have you considered how to frame this approach in the international standards regarding LCA methodology?

Reviewer #2: The paper is well organized and well written. The aim of the work is clear and coherent with the text content. I only few minor aspects to underline.

1) It might be useful for the reader to have a slightly more detailed description of the outputs of the model and of the procedure are and how they can be used in decision-making or programming.

2) Another aspect that is not clear to me is how the "weights" relating to the three pillars, i.e. environmental, economic and social, are attributed. These weights play a fundamental role because by changing the set of weights, you probably get very different scenarios of excellent performance. please provide some further details on this aspect.

3) It could be useful for the reader to have the possibility to have the application of the method to a real application case. Furthermore, I have the feeling that, due to the way the system is set up, decisions that minimize environmental impacts exert a greater influence on the model than the decision-making choices deriving from economic and social ecosystems. If this were the case, the method would essentially be simplified into a procedure capable of optimizing decisions to minimize environmental impacts, having decisions related to economic and social parameters a lower weight.

6. PLOS authors have the option to publish the peer review history of their article (what does this mean?). If published, this will include your full peer review and any attached files.

Reviewer #1: No

Reviewer #2: **Yes: **PROF. MARCO BOVO

---

## [Author Response · Author response to Decision Letter 0]

1 Dec 2024

Responses to the Editor

Comment: Dear Authors, Thank you for your submission to PLOS ONE. After careful and extensive review, I note that your submission requires major revisions before it can be accepted for publication. I invite you to consider all the review comments appended below and revise your manuscript to acceptable standards. I suggest to pay particular attention on LCA methodology and model validation as requested by Reviewer #1

Response: Thank you for the constructive feedback and for the opportunity to revise our manuscript. We have thoroughly reviewed and addressed each of the reviewer comments. We are confident that these revisions have improved the quality and clarity of the manuscript.

We would like to clarify a point in the Reviewer Responses to Questions, specifically regarding the third question: “Have the authors made all data underlying the findings in their manuscript fully available?” Reviewer 1 indicated “Yes”, while Reviewer 2 responded “No”. We apologize for any misunderstanding concerning data availability. All data and model codes used in this study are fully accessible and digitally available on GitHub, as referenced in [61] in the original manuscript and [62] in the revised manuscript. This availability was explicitly indicated upon submission and is detailed in multiple places in the manuscript (in both the original and revised versions), particularly in Section Computational experiments and discussion, including: 

‘..All the data used and the model codes are fully provided and digitally available on GitHub [61]/[62].’

‘..The LCA data structure adoption of the updated dataset is available in [61]/[62].’

‘..The scaling factors for the processes that are used in the solutions (greater than zero) are provided in S1 File and are available in [61]/[62].’

Responses to the comments of Reviewer #1

Comment no.1: Dear authors, the work is really interesting and innovative, which is properly stressed in the article. However, I think some aspects should be clarified before considering it for publication.

Response no.1: We appreciate the Reviewer’s positive feedback on the innovation in our work. We also thank the Reviewer for the thoughtful comments, which we believe we have strengthened our manuscript.

Comment no.2: Considering the number of acronyms, I suggest including a nomenclature section

Response no.2: Thank you for this suggestion. We agree and have added a table of acronyms to the Supporting information in the revised manuscript. This table (labelled as Table 1 in S1 File, Supplementary Material) is followed by Table 2, which lists symbols frequently used in the SCLCO model.

Comment no.3: The authors talk about integrating the LCA method at the level of supply chain optimization. An important feature of an LCA study is the definition of the impacts that a process can have on the environment. How is this aspect quantifiable in your integrated model? Do you foresee factors for characterizing the impacts and how are they integrated at the methodological level? In analogy to this aspect how is treated the stage of Environmental impact assessment?

Response no.3: We thank the Reviewer for emphasizing the significance of impact assessment in LCA, as it provides weights for emissions to the considered impact categories. While developing a specific impact assessment methodology is beyond this study’s scope, our model is designed to be compatible with established life cycle impact assessment (LCIA) methodologies (e.g., ReCiPe). As such, any standard impact assessment method can be applied to construct the characterization matrix Qenv in our SCLCO model. The use of different methodologies does not alter the model’s structure as it only generates different input for matrix Qenv. Based on the reviewer’s comment, we have clarified this point in Section Model formulation of the revised manuscript by the following paragraph (changes in blue): 

“System-wide aggregated environmental impacts are determined with the scaling vector, s. Environmental flows (represented in Benv) can be aggregated into high-level impact categories (e.g. global warming, ecotoxicity) during the “impact assessment” phase of LCA, also known as LCIA ([1]). In this phase, contributions of the environmental flows (e.g. CO2, CH4, SO2) are converted into quantified environmental impact categories (e.g. global warming, acidification) based on the conversion factors defined by an impact assessment method (e.g [51, 52]). This phase allows for both midpoint and endpoint methods. The characterization matrix Qenv includes the relative importance of the environmental flows (represented in Benv) for the selected life cycle impact categories. Qenv is a C×B-dimensional matrix, where C is the number of impact categories. The entries in Qenv may vary depending on the impact assessment methodology, and the number of rows (which corresponds to the number of impact categories addressed) also varies per impact assessment methodology.” 

Comment no.4: The methodology has been presented at the methodological and formulation level but it is not clear whether the model has ever been validated? Has it been tested on any real cases? This aspect is fundamental for the reliability and repeatability of the work.

Response no.4: We thank the Reviewer for emphasizing the importance of validating our model. The model has been tested against a traditional Supply Chain Optimization (SCO) model and a case study from the literature as part of our computational experiments. As detailed Section Computational experiments and discussion, we obtained results consistent with the traditional SCO model, confirming that our model can be used as an effective alternative. Within Section Computational experiments and discussion, we discuss the alternative outputs generated by our model, which provide a broader and more interpretable perspective, particularly for practitioners and researchers with an LCA background.

We also recognize the importance of a fully fledged real-case industry implementation, and we have highlighted this as one of the future research directions in Section Conclusions. While the model has not yet been directly implemented in a real-world industry setting, we are pleased to note that the case study presented in the manuscript mirrors a real-world electronic components supply chain (as emphasized by Tautenhain et al., 2021) signaling the potential for practical implementation. The case incorporates case-specific constraints (e.g., technology selection, minimum return fraction) and network structures. We have indicated the use of the case study and its background at the beginning of Section Computational experiments and discussion in the original manuscript, but regret that it was insufficiently clear. We have therefore tried to clarify the validation details in the revised manuscript, in Section Computational experiments and discussion by adding the following paragraph:

“The selected case, mirroring a real-world electronic components supply chain, incorporates case-specific constraints (e.g., technology selection, minimum return fraction) and network structures. The computational experiments will demonstrate that using the SCLCO model yields results consistent with the original SCO model, confirming that our model can be used as an effective alternative.”

Comment no.5: Have you considered how to frame this approach in the international standards regarding LCA methodology?

Response no.5: We thank the Reviewer for raising the question regarding alignment with international LCA standards. This alignment is one of the main concerns of our manuscript, as we aim to present an SCO model that calculates resulting environmental impacts based on the ISO-standardized LCA methodology. We have ensured that our methodology is compatible with ISO 14040 series, which guide the principles and framework for LCA. Key components of our model align with these standards and were discussed in the original manuscript in separate subsections, with supporting academic references: the matrix-data representation in Section Organizing data, the traditional LCA structure in Section Model formulation, the integration of existing LCA-based optimization models in Section Supply chain life cycle optimization (SCLCO), and the formulation of our final model in Section SCLCO formulation. Although we have indicated the model’s reliance on LCA and its compatibility with ISO standards in various parts of the original manuscript, we acknowledge that this connection may not have been as clear as intended. To explicitly reflect on the issue raised by the reviewer, we have added the following paragraph at the beginning of Section Methodology, 

“Our methodology is compatible with the international standards regarding LCA methodology (i.e., ISO 14040 series). Key components of our model align with these standards and are discussed in separate subsections, with supporting academic references: the matrix-data representation in Section Organizing data, the traditional LCA structure in Section Model formulation, the integration of existing LCA-based optimization models in Section Supply chain life cycle optimization (SCLCO), and the formulation of our final model in Section SCLCO formulation.”

and at the end of Section Model formulation,

“Breaking down Eq (2), we can clearly identify the three key LCA stages (defined by ISO [1]) from right to left, “inventory analysis” (Benvs), “impact characterization” (QenvBenvs), and “weighting” (wenvQenvBenvs). ”

Responses to the comments of Reviewer #2

Comment no.1: The paper is well organized and well written. The aim of the work is clear and coherent with the text content. I only few minor aspects to underline.

Response no.1: We thank the Reviewer for the positive feedback on our work. We appreciate the constructive comments provided and have addressed the minor points noted in the revised manuscript.

Comment no.2: It might be useful for the reader to have a slightly more detailed description of the outputs of the model and of the procedure are and how they can be used in decision-making or programming.

Response no.2: We thank the Reviewer for emphasizing the importance of clarifying the model outputs for readers. In the original manuscript, we have described the high-level decision variables of the SCLCO model in Section Decision variables and the modeling approach at the beginning of Section Supply chain life cycle optimization (SCLCO). We agree with the Reviewer that an additional concise explanation of the model outputs for the case study would enhance understanding. To address this, we have added the following explanation to the revised manuscript at the beginning of Section Interpretting the solutions of SCLCO: 

“s0 represents scaling factors for processes related to the strategic decisions and determines fleet composition, entity construction, and, technology and supplier selection. st includes scaling factors for processes related to the tactical decisions and defines raw material acquisition amounts, production and remanufacturing quantities, storage levels, and transportation choices (e.g., air or sea). Unlike traditional SCO models that output explicit flows, our model implicitly defines flows through s0 and st, as they represent the in- and outflows of the underlying processes.”

Comment no.3: Another aspect that is not clear to me is how the "weights" relating to the three pillars, i.e. environmental, economic and social, are attributed. These weights play a fundamental role because by changing the set of weights, you probably get very different scenarios of excellent performance. please provide some further details on this aspect.

Response no.3: 

The SCLCO model formulated in Equations (4a)-(4h) is multi-dimensional by nature, similar to Tautenhain et al., (2021). Given that we wanted to test our model on their case study, we followed their approach of optimizing each objective function separately in order to compare the results of both modelling approaches. As such, we do not impose specific weights each of the pillars, although it is of course possible to do so, if one wants to solve a weighted objective function.

To ensure that the reader fully understand our objective function is multi-objective, incorporating separate environmental, economic, and social objectives, we have added the following sentence to Section SCLCO formulation of the revised manuscript:

“ The multi-objective function is defined in Eq (4a). It includes separate environmental, economic, and social objectives. In other words, we do not prioritize one objective over the others.”

In our study, however, weights are used to assign relative importance to the different impact categories within each of the three pillars (environmental, economic, and social), which aligns with the “weighting” step of LCA. Specifically, we define three weight vectors (wenv, wecn, wsoc), each representing the weights for the impact categories within its respective pillar. These weights are inputs to the model. The issue of assigning weights is a topic of ongoing debate (see, e.g., Bayazıt et al. ,2024), but our approach focuses on decision-making given a predefined set of weights, rather than determining the weights themselves. To clarify this, we have added the following sentence to the revised manuscript in Section Model formulation:

“We acknowledge the issue of weighting is topic of an ongoing debate (see, e.g., [53]). Nevertheless, given a vector of weighting factors, the proposed method works smoothly.”

Comment no.4: It could be useful for the reader to have the possibility to have the application of the method to a real application case. 

Response no.4: We thank the Reviewer for the comment. The case study presented already draws from a real-world application within the electronic components supply chain, providing an illustrative yet realistic representation of the decision-making process in this context. We have added the following sentence in Section Computational experiments and discussion to make this clear (see also our response to Comment 4 of Reviewer 1): 

“The selected case, mirroring a real-world electronic components supply chain, incorporates case-specific constraints (e.g., technology selection, minimum return fraction) and network structures. The computational experiments will demonstrate that using the SCLCO model yields results consistent with the original SCO model, confirming that our model can be used as an effective alternative.” 

Comment no.5: Furthermore, I have the feeling that, due to the way the system is set up, decisions that minimize environmental impacts exert a greater influence on the model than the decision-making choices deriving from economic and social ecosystems. If this were the case, the method would essentially be simplified into a procedure capable of optimizing decisions to minimize environmental impacts, having decisions related to economic and social parameters a lower weight.

Response no.5: In its current form, our model is structured as a multi-objective framework that includes separate environmental, economic, and social objectives, ensuring that no objective is prioritized over another (please see our Response 3). The model/system is designed to comprehensively account for all relevant impacts across processes, minimizing the risk of overlooking any decision impacts-covering the impacts from the three pillar, economic, social and environmental.

In our computational experiments (Section Computational experiments and discussion), we solved a case study for each objective separately following the approach of Tautenhain et al. (2021) to compare the results of both modelling approaches. The choice of the objective function has a fundamental impact on the structure of the solutions obtained. For instance, the environmental solution favours remanufacturing, while the economic solution minimizes it, and the social solution increases truck purchases to create jobs, while the economic solut

---

## [Decision Letter · Decision Letter 1]

17 Dec 2024

Integrating Life Cycle Assessment into Supply Chain Optimization

PONE-D-24-36569R1

Dear Dr. Hulagu,

We’re pleased to inform you that your manuscript has been judged scientifically suitable for publication and will be formally accepted for publication once it meets all outstanding technical requirements.

Kind regards,

Alberto Barbaresi

Academic Editor

PLOS ONE

Additional Editor Comments (optional):

Reviewers' comments:

Reviewer's Responses to Questions

**Comments to the Author**

1. If the authors have adequately addressed your comments raised in a previous round of review and you feel that this manuscript is now acceptable for publication, you may indicate that here to bypass the “Comments to the Author” section, enter your conflict of interest statement in the “Confidential to Editor” section, and submit your "Accept" recommendation.

Reviewer #1: All comments have been addressed

Reviewer #2: All comments have been addressed

2. Is the manuscript technically sound, and do the data support the conclusions?

Reviewer #1: Yes

Reviewer #2: Yes

3. Has the statistical analysis been performed appropriately and rigorously? 

Reviewer #1: N/A

Reviewer #2: N/A

4. Have the authors made all data underlying the findings in their manuscript fully available?

Reviewer #1: Yes

Reviewer #2: Yes

5. Is the manuscript presented in an intelligible fashion and written in standard English?

Reviewer #1: Yes

Reviewer #2: Yes

6. Review Comments to the Author

Reviewer #1: Dear authors,

thank you for having addressed all the previous comments. No further comments have been considered before publication.

Reviewer #2: (No Response)

7. PLOS authors have the option to publish the peer review history of their article (what does this mean?). If published, this will include your full peer review and any attached files.

Reviewer #1: No

Reviewer #2: **Yes: **MARCO BOVO

---

## [Editor Report · Acceptance letter]

3 Jan 2025

PONE-D-24-36569R1 

PLOS ONE

Dear Dr. Hülagü, 

I'm pleased to inform you that your manuscript has been deemed suitable for publication in PLOS ONE. Congratulations! Your manuscript is now being handed over to our production team.

Kind regards, 

on behalf of

Dr. Alberto Barbaresi 

Academic Editor

PLOS ONE